# Vitamin D and Intrauterine Growth Restriction (IUGR)

**DOI:** 10.3390/ijms262311422

**Published:** 2025-11-26

**Authors:** Teodoro Durá-Travé, Fidel Gallinas-Victoriano

**Affiliations:** 1Department of Pediatrics, School of Medicine, University of Navarra, 31008 Pamplona, Spain; 2Navarrabiomed (Biomedical Research Center), 31008 Pamplona, Spain; fivictoriano@hotmail.com; 3Department of Pediatrics, Navarra University Hospital, 31008 Pamplona, Spain

**Keywords:** intrauterine growth restriction, low birth weight, pregnancy, small for gestational age, vitamin D deficiency, vitamin D supplementation

## Abstract

During pregnancy, the fetus is entirely dependent on maternal sources of vitamin D, which also regulates placental function. Vitamin D deficiency during pregnancy has been associated with intrauterine growth restriction (IUGR). This study aims to provide a narrative review of the potential influence of vitamin D deficiency on the pathogenesis of IUGR, and the potential benefits of vitamin D supplementation during pregnancy on fetal anthropometry. This review highlights the high prevalence of vitamin D deficiency among pregnant women and newborns worldwide, even in sunny countries. Most studies support that fetal vitamin D levels are directly related to maternal vitamin D levels. There is extensive literature confirming the relation between maternal vitamin D status and fetal growth patterns throughout pregnancy (both early and late). However, there is currently insufficient evidence to establish recommendations on optimal prenatal vitamin D supplementation in women to reduce the risk of IUGR.

## 1. Introduction

Vitamin D is known as the ‘sunshine hormone’ [1]. Its levels depend on various factors, including skin pigmentation, physical barriers that hinder the absorption of solar radiation (e.g., clothing and sunscreens), and geographical location (e.g., latitude, climate, season, and altitude) [2,3]. However, irrespective of its associated factors, vitamin D deficiency is now considered a worldwide pandemic representing a public health concern with multiple health consequences [4,5,6,7].

Apart from its fundamental role in regulating calcium and phosphorus homeostasis for skeletal development, vitamin D also mediates other functions, including cell proliferation and differentiation, anti-inflammatory action, and the regulation of innate and acquired immunity. Calcidiol, the primary circulating vitamin D metabolite, is widely recognized as the most reliable indicator of vitamin D status thanks to its long half-life, stability, and responsiveness to endogenous production or exogenous intake of vitamin D [8]. In fact, it is now commonly referred to as the “barometer” of vitamin D status. The American Endocrine Society criteria for the classification of vitamin D status [9] are currently the most widely accepted and used by authors (Table 1). However, while calcidiol is currently the most accepted biomarker for assessing vitamin D status, its optimal levels remain a matter of debate [10].

During pregnancy, the fetus is entirely dependent on the maternal sources of vitamin D, which also regulate placental function [11]. The metabolic physiology of pregnant women differs from that of non-pregnant women. Indeed, during pregnancy, the mother modifies her physiological and homeostatic mechanisms to satisfy the physiological needs of the growing fetus. In recent years, the potential impact of vitamin D deficiency during pregnancy on maternal and neonatal health has received increasing attention. Vitamin D is essential for placental function, calcium homeostasis and fetal bone mineralization—all of which are important for fetal development and growth [12,13,14]. Furthermore, as vitamin D plays an immunomodulatory role, it has been suggested that it contributes to maternal-fetal immune tolerance; without this, the fetus could not survive [2,15]. Therefore, it is important to maintain adequate maternal vitamin D levels during pregnancy to prevent adverse outcomes for the pregnancy, fetus and postnatal period. Pregnant women and newborns are recognized as populations at an increased risk of vitamin D deficiency [2,3,4,16,17,18,19]. Vitamin D deficiency during pregnancy has been associated with an increased risk of spontaneous early pregnancy loss, pre-eclampsia, gestational diabetes mellitus, cesarean section, bacterial vaginosis, postpartum depression, preterm delivery and low birth weight (LBW), and being small for gestational age (SGA) [7,20,21,22,23,24,25,26,27,28,29,30].

This review aims to provide a comprehensive narrative review of (a) recent data on the prevalence of vitamin D deficiency in pregnant women; (b) the relationship between maternal and neonatal vitamin D levels; (c) the possible influence of vitamin D deficiency on the pathogenesis of intrauterine growth restriction (IUGR); (d) the potential benefits of vitamin D supplementation during pregnancy on neonatal anthropometry. The review is based on an electronic literature search of the PubMed database of the US National Library of Medicine, performed by two independent researchers. This search covered publications from January 2010 to June 2025. The following Medical Subject Headings or keywords were used in the search, either on their own or in combination: ‘vitamin D’, ‘vitamin D deficiency’, ‘pregnancy’, ‘intrauterine growth restriction’, ‘small for gestational age’, ‘low birth weight’, and ‘vitamin D supplementation’.

## 2. Vitamin D Metabolism and Adaptive Changes During Pregnancy

Vitamin D (cholecalciferol) is a prohormone synthesized primarily endogenously in the skin under the influence of ultraviolet B radiation from the sun, and it requires double hydroxylation to become functional. It is then released into the bloodstream, where it is transported to the liver bound to vitamin D-binding protein (VDBP). Here, the enzyme cholecalciferol-25-hydroxylase catalyzes the first hydroxylation, creating 25-hydroxycholecalciferol, also known as calcidiol. Calcidiol is the primary circulating vitamin D metabolite and is used to determine vitamin D status. However, it is not the active form of vitamin D. Calcidiol is released into the bloodstream, where it is transported bound to VDBP to the kidneys. There, the enzyme 1α-hydroxylase (CYP27B1) catalyzes a second hydroxylation, converting calcidiol to 1,25-dihydroxyvitamin D_3_ (calcitriol), the biologically active form of vitamin D. CYP27B1 is primarily expressed in the kidneys, but is also found in multiple tissues, including the placenta.

Most of the circulating calcitriol is produced by renal CYP27B1. The main regulators of renal calcitriol synthesis are parathyroid hormone (PTH), which signals calcium serum levels, fibroblast growth factor (FGF23), which signals serum phosphate levels, and calcitriol itself. In fact, calcitriol regulates its own catabolism via feedback mechanisms, whereby high concentrations activate the functionality of the CYP24A1 enzyme (24-hydroxylase activity), which catalyzes the conversion of calcidiol and calcitriol into 24,25(OH)_2_D and 1,24,25(OH)_3_D (vitamin D metabolites with virtually no biological activity). This simultaneously reduces the enzymatic activity of CYP27B1, resulting in a decrease in calcitriol levels.

The role of vitamin D in the human body is not limited solely to regulating calcium levels and bone health. It has a variety of effects that extend beyond these areas. Indeed, vitamin D is now regarded as a pleiotropic hormone that operates via genomic and non-genomic pathways. Most of the effects of vitamin D are mediated by its interaction with the vitamin D receptor (VDR), which is a nuclear transcription factor that binds to specific DNA sequences. This modulates the expression of a plethora of target genes (approximately 5–10% of the total human genome) that are implicated in numerous physiological processes. These include cellular proliferation and differentiation, as well as anti-inflammatory and immunomodulatory activities (genomic pathway). Additionally, vitamin D induces rapid, non-genomic cellular responses by binding to a membrane receptor called `membrane-associated rapid response steroid’ in order to regulate several intracellular processes [2,22,31,32,33,34,35].

During pregnancy, significant changes occur in maternal calcium metabolism to support adequate fetal bone mineralization. Consequently, several physiological adaptations occur, including increased maternal serum calcitriol and VDBP, increased placental VDR and renal and placental CYP27B1 activity, and decreased calcitriol catabolism [24,28]. Figure 1 shows the main adaptive changes in vitamin D metabolism during pregnancy.

In pregnant women, serum calcitriol levels increase from the first trimester, tripling by the end of the third trimester compared to the preconception stage (calcitriol does not cross the placental barrier). This is a consequence of the significant increase in maternal renal synthesis of calcitriol during pregnancy, which is due to increased CYP27B1 activity. Furthermore, the placenta also plays a role, as both the maternal (decidual) and fetal (trophoblastic) placentas exhibit CYP27B1 activity and, as extra renal organs, contribute to vitamin D activation. Increased maternal calcitriol levels would promote intestinal calcium absorption and consequently a secondary increase in plasma levels to meet fetal calcium demands for skeletal mineralization through transplacental passage [36]. Other hormones such as estradiol, prolactin and placental lactogen, which increase substantially during pregnancy, can contribute to upregulate intestinal calcium absorption. Calcium absorption exceeds maternal-fetal needs, which is generally compensated by increased renal calcium excretion that reaches the hypercalciuric range (absorptive hypercalciuria). The mechanisms that lead to an increase in CYP27B1 activity during pregnancy remain unclear, partly because the known regulatory factors, such as parathyroid hormone (PTH), remain suppressed during gestation (PTH as a marker of vitamin D status is unreliable during pregnancy). It has been hypothesized that the PTH-related peptide (PTH-rP) could be a potential regulatory factor for CYP27B1 activity, in conjunction with other factors (placental lactogen, estradiol, and prolactin). This peptide is a PTH analog that increases throughout pregnancy and is synthesized in the breast and placenta (both maternal and fetal), and probably also in the fetal parathyroid glands. Its activity could largely explain the significant increase in calcitriol and the suppression of PTH levels during pregnancy [37,38]. Furthermore, PTHrP appears to stimulate the placental transfer of calcium and other minerals.

**Figure 1 ijms-26-11422-f001:**
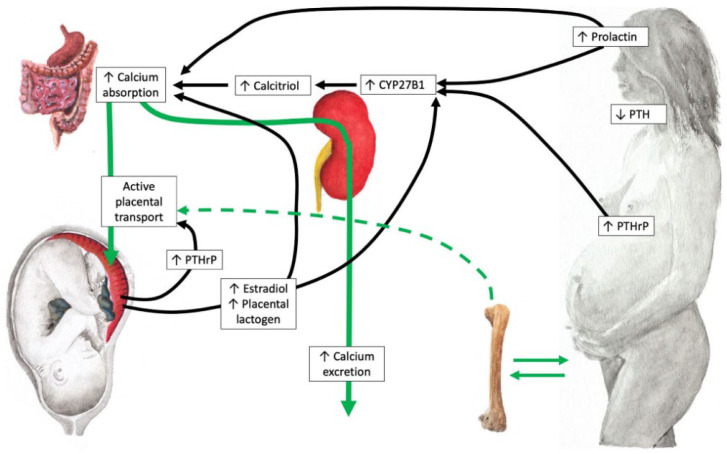
Maternal mineral and bone physiology during pregnancy (reproduced with permission from Ryan et al. [38]). The main maternal adaptation is a doubling in fractional intestinal absorption of calcium and other minerals, providing the baby with what it needs. This increase is partly driven by a two- to threefold increase in calcitriol, but other factors (prolactin, placental lactogen, PTHrP, and likely others) can also upregulate intestinal calcium absorption in the absence of calcitriol. The production of calcitriol in the kidneys is not stimulated by parathyroid hormone (PTH), which is suppressed during pregnancy, but rather by factors such as high levels of estradiol, PTHrP, prolactin, placental lactogen and possibly other factors. In addition, PTHrP stimulates placental calcium transfer. Typically, more calcium is absorbed than is needed, leading to increased renal calcium excretion that reaches the hypercalciuric range (absorptive hypercalciuria). Green arrows represent calcium flow pathways, while black arrows represent regulatory pathways.

Maternal VDBP levels increase significantly during pregnancy. Due to VDBP’s greater affinity for calcidiol than calcitriol, some authors have suggested that this increase may act as a reservoir for calcidiol and/or redistribute it into the fetal circulation. As previously mentioned, in non-pregnant women, calcitriol regulates its own catabolism through feedback mechanisms.

However, during pregnancy, placental methylation of the CYP24A1 enzyme reduces its functionality. Consequently, the activation of the vitamin D catabolism mechanism is reduced. This epigenetic imbalance in the vitamin D feedback catabolite greatly contributes to the increase in maternal calcitriol levels during pregnancy. This ensures greater calcium transfer from mother to fetus [2,39].

One of the most important functions of vitamin D during pregnancy is to increase maternal calcium absorption and placental transport. However, as VDR and CYP27B1 are also expressed in other female reproductive tissues, including the uterus, ovaries, endometrium and fallopian tubes, the possibility of other paracrine and autocrine actions of calcitriol cannot be ruled out. A critical role is also played by vitamin D in regulating the innate and adaptive immune systems, which is crucial since an adequate balance of cytokines is necessary for a successful pregnancy. Vitamin D enhances the antimicrobial properties of epithelial barriers by regulating the genetic expression of potent antimicrobial peptides, such as cathelicidin and β-defensins, which destroy the cell membranes of bacteria and viruses. Additionally, vitamin D inhibits the production of inflammatory cytokines, including interleukin (IL)-1, IL-2, IL-6, IL-8, IL-12, IL-17 and IL-21, as well as tumor necrosis factor-alpha (TNF-α) and interferon-gamma. It also increases the production of anti-inflammatory cytokines: IL-4, IL-5, IL-10 and transforming growth factor beta. In other words, vitamin D plays a dual role of improving the innate immune response and neutralizing potentially exacerbated inflammation [22,40,41,42]. The biological effects of vitamin D on the placenta are also related to hormonogenesis and overall placental physiology, which are both important factors in maternal and fetal health. Vitamin D induces endometrial decidualization and the synthesis of estradiol and progesterone; however, it also regulates the expression of human chorionic gonadotropin and placental lactogen. Given the significant impact of vitamin D on human pregnancy, it is not surprising that vitamin D deficiency could contribute to various pregnancy-related disorders [2,24].

## 3. Prevalence of Vitamin D Deficiency in the Pregnant Women

As previously mentioned, various factors influence vitamin D status, which should be considered when comparing the prevalence of vitamin D deficiency in countries that differ in terms of ethnicity, culture or geography [2,3]. Vitamin D deficiency has been reported among pregnant women worldwide. For instance, a 2013 narrative review indicated that, although the prevalence of vitamin D deficiency during pregnancy varies by geographic area of residence, it is a particular global concern. Indeed, the review found that vitamin D deficiency in pregnant women ranged from 27% to 91% in the United States, 39% to 65% in Canada, 70% to 100% in Northern Europe, 78% to 100% in the Middle East, 45% to 98% in Asia, and 25% to 65% in Australia. The review provided a global overview of the incidence of vitamin D deficiency in pregnant women worldwide, and the available data indicated that vitamin D deficiency is a global public health issue affecting all age groups, particularly pregnant women, even in countries with year-round sun exposure [2].

A systematic review with meta-analysis was subsequently published in 2016, aiming to create a global summary of maternal and neonatal vitamin D status [16]. The review involved pregnant women and their newborn babies from different population groups corresponding to WHO regions for which data was available (North America, Europe, the Eastern Mediterranean, South-East Asia and the Western Pacific). It indicates that vitamin D deficiency was present in 54% of pregnant women and 75% of newborn babies. Severe deficiency was present in 18% of pregnant women and 29% of newborns. There was some variability among the different geographic areas included in the study. The regional ranking of the proportion of pregnant women with vitamin D deficiency was as follows: South-East Asia: 87%, Western Pacific: 83%, Americas: 64%, Europe: 57%, and Eastern Mediterranean: 46%. The regional ranking of the proportion of newborns with vitamin D deficiency was as follows: South-East Asia: 96%; Europe: 73%; Eastern Mediterranean: 60%; Western Pacific: 54%; and the Americas: 30%. A systematic review and meta-analysis involving participants from 23 African countries is noteworthy with regard to the African continent. Despite significant methodological heterogeneity, the meta-analysis revealed that the prevalence of vitamin D deficiency in the general African population, particularly among pregnant women (44%) and newborns (49%), was higher than expected given the continent’s abundant sunshine. This was mainly observed in populations living in North Africa and South Africa, as opposed to sub-Saharan Africa [43].

Recently, several studies have been published in countries located at low latitudes, which are assumed to have sufficient solar radiation to prevent vitamin D deficiency. However, these studies concluded that exposure to abundant sunlight in these countries is insufficient to prevent hypovitaminosis D, and therefore recommend that pregnant women take vitamin D supplements and consume foods fortified with vitamin D. For instance, one observational study examined the prevalence of vitamin D deficiency among pregnant women at the Shanghai Changning District Maternal and Child Health Hospital. A total of 34,417 women were included in the study, and their vitamin D status was evaluated in the 16th of gestation. The results revealed that 98.4% of pregnant women presented with hypovitaminosis D (28.4% had insufficient levels and 70% had deficient levels), while only 1.6% of participants had adequate levels of vitamin D [44]. In contrast, a cross-sectional study conducted in the Obstetrics department of a tertiary care hospital in Chennai, India, estimated a prevalence of vitamin D deficiency of 62% among pregnant women in their third trimester. Linear regression analysis showed that sun exposure was a significant predictor of serum calcidiol levels among antenatal mothers [45]. Additionally, a prospective cohort study was conducted at the KK Women’s and Children’s Hospital in Singapore. A total of 93 pregnant women in their first trimester (with a gestation period of over 15 weeks) were recruited and their vitamin D status was assessed. Only 2.2% of participants had sufficient vitamin D levels; the rest had hypovitaminosis D (insufficient: 49.5%; deficient: 48.4%) [46]. A systematic review and meta-analysis of cross-sectional and observational studies of pregnant women in Indonesia has been conducted. This study examined 830 pregnant women and found that 78% had hypovitaminosis D (25% had insufficient levels and 63% had deficient levels) [47]. Both Singapore and Indonesia are equatorial countries located in Southeast Asia with no seasonal changes. While ethnic and/or cultural factors, such as higher skin pigmentation and traditional dress like the hijab or shari, have been suggested as possible explanations, the very high prevalence of hypovitaminosis D in such sunny countries remains an enigma and a public health emergency.

Recently, several observational studies have been conducted concerning the prevalence of vitamin D deficiency in pregnant women in countries with different ethnic and geographic characteristics. Table 2 shows the results of some of these studies.

## 4. Relationship Between Maternal and Neonatal Vitamin D Levels

Much recent research has focused on the relationship between maternal and neonatal vitamin D levels. Studies have indicated an association between newborn vitamin D levels at birth and those of their mothers. Most studies support the idea that fetal vitamin D concentrations depend on maternal concentrations [58,59,60,61,62].

One notable study is a prospective, multiethnic, population-based cohort study of 7098 mother-child pairs in Rotterdam, the Netherlands, conducted as part of the Generation R Study. Maternal blood samples were collected in the second trimester of pregnancy, and umbilical cord blood samples were collected after delivery. A statistically significant correlation (r = 0.62) was found between maternal vitamin D concentrations in the second trimester and umbilical cord blood [12]. Several recent studies corroborate the relationship between maternal and neonatal vitamin D levels. For example, an observational study conducted at the Obstetrics Clinic of the County Clinical Hospital in Târgu Mureș, Romania, included 131 mothers and their newborns at the time of delivery (37–42 weeks of gestation). All of the mothers in the study had vitamin D insufficiency (26%) or deficiency (49%). The results clearly showed a positive linear correlation between maternal and neonatal serum vitamin D levels (r = 0.96, *p* < 0.01) and emphasize the importance of investigating the impact of vitamin D deficiency on neonatal health outcomes [55]. Another descriptive-observational study was conducted with 102 mother-child pairs at a tertiary care center (Hind Institute of Medical Sciences, Barabanki, India). All of the included mothers had given birth to healthy, full-term babies. The study found that 65.7% of mothers and 78.4% of babies had vitamin D deficiency. There was also a clear link between the vitamin D levels of the mothers and babies (r = 0.68; *p* < 0.0001). The authors highlight that even in regions with abundant sunlight, pregnant women and their newborns had inadequate serum vitamin D concentrations [56].

Finally, it is worth mentioning a cross-sectional study conducted on 248 neonates and their mothers at Tzaneio General Hospital in Piraeus, Greece (a country with low latitudes where ultraviolet radiation is generally assumed to be sufficient to prevent vitamin D deficiency). The vitamin D status of the mothers before delivery and of the neonates in their umbilical cord blood was studied. Vitamin D deficiency was found in 58% of mothers, while 25% were vitamin D-insufficient. Only 17% of pregnant women had normal vitamin D status. Among newborns, vitamin D deficiency was recorded in 66% of cases and vitamin D insufficiency in 29%. Only 5% of neonates had normal vitamin D status. Additionally, a strong direct correlation was observed between maternal and neonatal vitamin D concentrations with a Pearson correlation coefficient of 0.8 [57]. Similar results were obtained in a retrospective study conducted at a tertiary maternity hospital in Bucharest, Romania, involving 130 pregnant women and their newborns. A significant, strong, positive, direct correlation (r = 0.79) was observed between maternal and neonatal vitamin D status [58]. One of the most interesting findings of both studies was that most newborns of mothers with severe vitamin D deficiency also presented with deficiency. Additionally, an analytical cross-sectional study involving 150 normal pregnant women in labor at term (>37 weeks of gestation) at a tertiary center in the Bundelkhand region of India was published just a few months ago. Maternal and infant blood samples were obtained at the time of delivery. The results revealed a positive correlation between serum vitamin D levels in newborns and maternal vitamin D levels. This finding supports the idea that normal vitamin D status in pregnant women is necessary for the fetus. If the mother is deficient, the same condition occurs to the fetus [59]. In other words, fetal and/or neonatal serum calcidiol levels are directly related to maternal calcidiol levels. Therefore, vitamin D deficiency in pregnant women could lead to decreased maternal calcium absorption and, consequently, reduced transfer to the placenta.

## 5. Maternal Vitamin D Status and Risk of Intrauterine Growth Restriction

IUGR (Intrauterine Growth Restriction) is defined as a fetus’s inability to achieve its genetic growth potential. This can be caused by maternal, placental, fetal, or environmental factors. IUGR is commonly diagnosed by ultrasound during pregnancy or by a detection of fetal weight below the 10th percentile for sex and gestational age at birth (SGA). LBW is defined as a birth weight of less than 2500 g, regardless of gestational age. Numerous observational studies, systematic reviews, and meta-analyses have demonstrated an association between lower maternal vitamin D levels and adverse fetal growth outcomes, including LBW, shorter bone length, and SGA births [12,21,51,63,64,65,66,67,68]. However, other studies have found no influence of vitamin D status on fetal anthropometric parameters [17,19,49,69,70,71].

The biological basis of the association between maternal vitamin D status and fetal growth patterns remains unclear. Normal placental development requires implantation of the blastocyst. This process begins with the embryo attaching to the maternal endometrial epithelium. Then, the fetal trophoblast invades the maternal endometrium. This enables the supply of oxygen and nutrients to the fetus and the excretion of waste product. At the same time, the process protects the fetus from maternal immune attack. Furthermore, the presence of VDR and CYP27B1 in placentas from early pregnancies suggests that vitamin D plays an important role in placental physiology. Both maternal decidua and fetal trophoblasts (including the syncytiotrophoblast and invasive extravillous trophoblast) express CYP27B1 and synthesize calcitriol. The coincident expression of VDR in the trophoblast and the decidua indicates that vitamin D synthesized locally could have an autocrine or paracrine function in the placenta. Ex vivo and in vitro analyses of trophoblast cells indicate that vitamin D plays a significant role in placental function. This includes the regulation of trophoblast differentiation and the invasion of the decidua and myometrium by invasive extravillous trophoblasts. Indeed, vitamin D appears to regulate genes responsible for trophoblast invasion and angiogenesis, both of which are essential for implantation, placental function and, consequently, fetal growth [72]. In short, vitamin D deficiency could lead to abnormal placentation and cause significant pregnancy disorders such as spontaneous abortion, preeclampsia, and fetal growth restriction [73].

On the other hand, vitamin D appears to play an important role in the growth and mineralization of the fetal skeleton. Skeletal formation begins during the embryonic period, but the primary period of skeletal mineralization occurs in the third trimester [37,38]. This period of fetal development is characterized by rapid growth, mineralization, and high serum mineral concentrations (calcium and phosphate). The placenta must supply the necessary minerals for adequate fetal skeletal mineralization by actively transporting calcium and phosphorus from the maternal circulation. Intrauterine skeletal mineralization is primarily determined by plasma ionic calcium concentration, which depends on placental calcium transfer and fetal calciotropic hormones, such as PTHrP, calcitriol, FGF23, calcitonin, and sex steroids. Active calcium transport across the placenta is regulated by the expression of plasma membrane calcium-dependent ATPases (PMCA1-4). Furthermore, PMCA3 gene expression predicts neonatal whole-body bone mineral content at birth. There is also experimental evidence that the expression of the PMCA gene is mediated by calcitriol [74]. In fact, serum calcium and phosphorus concentrations are higher in the fetus than in the mother, facilitating the mineralization of the developing skeleton. In adults, such high mineral content would lead to serious consequences, including soft tissue calcification, coronary artery calcification and calciphylaxis. Figure 2 depicts the sources of minerals during fetal development. Vitamin D deficiency in pregnant women is likely to be associated with maternal hypocalcemia. If placental calcium transfer is deficient, compensatory fetal hyperparathyroidism will occur because maternal PTH does not cross the placenta. This leads to calcium reabsorption from the fetal skeleton, resulting in decreased bone mineral content and potential fetal growth restriction [37,38].

Despite their varied designs, we would like to highlight a series of recent studies that are methodologically rigorous. We consider these studies worthy of mention because they provide novel scientific evidence in this field. The SCOPE (Screening for Pregnancy Endpoints) study was a large, international, prospective pregnancy cohort study involving 1786 women at 15 weeks of gestation. The study reported an inverse association between calcidiol concentrations greater than 30 ng/mL and uteroplacental dysfunction, as indicated by a composite outcome of SGA and preeclampsia. In other words, vitamin D status was associated with uteroplacental dysfunction, as indicated by preeclampsia and SGA birth [75].

A recent experimental study using mouse models examined the underlying mechanism by which gestational vitamin D deficiency induces IUGR. The female mice were divided into vitamin D-deficient and control groups. The results showed that gestational vitamin D deficiency induced IUGR (the fetal weight and length were lower in the deficient mice than in the controls) and inhibited placental development (the placental weight and diameter were lower in the deficient mice than in the controls). Additionally, gestational vitamin D deficiency was found to impair placental function, as evidenced by severe damage to the placental labyrinth layer and a reduction in the internal space of the placental vessels in vitamin D-deficient mice. Furthermore, gestational vitamin D deficiency caused placental inflammation, as demonstrated by higher levels of inflammatory cytokines and chemokines, including placental TNF-α, MCP-1, and KC mRNA, in the maternal sera of subjects with vitamin D deficiency compared to controls. Thus, these findings provide experimental evidence that gestational vitamin D deficiency leads to placental insufficiency and IUGR, potentially through the induction of placental inflammation [76]. More recently, an experimental study in rats showed that IUGR was significantly more prevalent in pregnant rats with vitamin D deficiency. Placental structure and function were impaired, and there was a clear inflammatory response accompanied by a significant increase in the phosphorylation level of the transcriptional coactivator Yes-associated protein (YAP). In other words, maternal vitamin D deficiency leads to placental dysplasia and IUGR, and these abnormal changes may be linked to the activation of the Hippo-YAP signaling pathway [77]. Therefore, adequate vitamin D supplementation (cholecalciferol) could be a potential strategy to prevent adverse pregnancy outcomes, particularly IUGR, in deficient pregnant women.

A prospective cohort study was conducted within the Eunice Kennedy Shriver National Institute of Child Health and Human Development Fetal Growth Studies–Singletons. Three hundred and twenty-one mother-offspring pairs were recruited from 12 clinics across the US. The study aimed to analyze the association between maternal vitamin D status during pregnancy and neonatal anthropometry at birth. Maternal vitamin D concentrations were measured at 0–14, 15–26, 23–31 and 33–39 weeks of gestation, and neonatal anthropometric measures were taken after delivery. The association between maternal vitamin D status and neonatal anthropometry varied throughout pregnancy. Vitamin D deficiency at 10–14 gestational week was associated with lower birthweight and shorter length, at 23–31 gestational week was associated with shorter length and lower sum of skinfolds, and at 33–39 gestational week, with shorter length. If these findings were confirmed, it would be important to monitor vitamin D status during pregnancy, since vitamin D deficiency at both the beginning and end of pregnancy could affect fetal growth [78].

The GraviD study is a prospective cohort study conducted in the Västra Götaland region of Sweden. The study included a total of 1810 mother-child pairs and aimed to examine the association between vitamin D status trajectories during pregnancy and newborn size at birth. Maternal vitamin D status was measured using samples taken from each participant at two time points: the first trimester (8–12 weeks of gestation) and the third trimester (32–35 weeks of gestation). Lower vitamin D status in early pregnancy was associated with pregnancy loss. Additionally, higher vitamin D status in late pregnancy, but not early pregnancy, was associated with a lower probability of having a baby that was SGA or had LBW. In other words, vitamin D status in the third trimester of pregnancy appears to be a better predictor of fetal growth restriction than in the first trimester [79].

A recent retrospective, population-based study was conducted at the Maternal and Child Health and Family Planning Service Center in Tumen City, Yanbian, China. The study included 510 pairs of healthy mothers and newborns. The study aimed to evaluate the correlation between maternal vitamin D deficiency and neonatal weight, as well as the predictive value of maternal vitamin D status in relation to fetal weight. Maternal vitamin D levels were measured at 16–20 weeks of gestation, and it was found that mothers with vitamin D deficiency were at an increased risk of delivering a baby that was SGA or had LBW. In other words, mid-pregnancy vitamin D levels could predict neonatal birth weight. Furthermore, receiver operating characteristic curve analysis indicated that vitamin D status could predict neonatal birth weight [80].

In the large, population-based, prospective cohort study integrated into the Generation R Study, maternal blood samples were collected in the second trimester of pregnancy (median gestational age: 20.3 weeks) and fetal growth measurements were performed in the second and third trimesters (median gestational ages: 20.5 and 30.3 weeks, respectively) using ultrasound procedures. The results showed an association between maternal vitamin D concentrations during pregnancy and longitudinally measured fetal head circumference, length, and weight. These findings suggest that lower maternal vitamin D concentrations in the second trimester are associated with restricted fetal head, length, and weight growth in the third trimester, as well as an increased risk of LBW and/or SGA at birth. Clearly, these results support the use of vitamin D supplementation during pregnancy to normalize vitamin D status [12].

The results of the aforementioned analytical cross-sectional study, which was carried out in 150 mothers and their newborn babies in the Bundelkhand region of India, are remarkable. Newborns of mothers with maternal vitamin D deficiency were 96% more likely to have LBW (i.e., <2500 g) than newborns of mothers with sufficient vitamin D levels. (i.e., >2500 g). In short, a significant, strong, positive, direct correlation (r = 0.83) was observed between maternal vitamin D status during pregnancy and birth weight [53].

Just a few months ago, a secondary analysis was published using data and samples from a multicenter prospective cohort study of nulliparous pregnant women conducted in the United States (the Nulliparous Pregnancy Outcomes Study). Vitamin D status was measured in 351 participants at 6–13 and 16–21 weeks of gestation. Fetal growth was measured by ultrasound at 16–21 and 22–29 weeks of gestation, and neonatal anthropometric measurements were performed at birth. The results showed that first-trimester maternal vitamin D status was positively associated with fetal linear growth but not with growth patterns for weight or head circumference. However, second-trimester vitamin D status was not associated with fetal growth patterns. The authors suggest that vitamin D supplementation should begin in the preconceptional period, and that future research should investigate the mechanisms by which vitamin D contributes to fetal growth [30].

## 6. Effects of Vitamin D Supplementation During Pregnancy on Birth Size

Given the high prevalence of vitamin D deficiency among pregnant women and the potential consequences for maternal and fetal health, it seems risky to rely exclusively on sun exposure and vitamin D-enriched foods to achieve adequate levels [81]. Several studies have evaluated the benefits of vitamin D supplementation during pregnancy. While vitamin D (cholecalciferol) supplementation may increase calcidiol levels in both mother and infant, it is unclear if it protects against intrauterine growth restriction due to the heterogeneity in the design of the study, the type of intervention, and the observational study designs [27,63,82,83,84,85]. Additionally, there is no universal agreement on the appropriate vitamin D dosage during pregnancy. For instance, the World Health Organization recommends a dosage of only 200 IU per day for pregnant women diagnosed with vitamin D deficiency. The UK National Institute for Health and Clinical Excellence recommends 400 IU/day [86]. The Food and Nutrition Board at the Institute of Medicine of the National Academies recommends 600 IU per day [87], while the U.S. Endocrine Society recommends 1500–2000 IU per day [9]. However, historical vitamin D supplementation trials suggest that 400 IU/day of vitamin D during pregnancy is grossly inadequate. Studies have shown that supplementation of up to 4000 IU per day is more effective in reducing deficiency and improving serum calcidiol levels in pregnant women and their neonates than 2000 IU or 400 IU per day, with no adverse maternal or fetal effects [8,84,88,89,90].

Inconsistent findings across studies could result from residual confounding factors. Maternal vitamin D status depends on diet, nutrition, vitamin supplementation, physical activity, socioeconomic status, and race, all of which can influence pregnancy outcomes independently. The contrasting results obtained in these studies have led several researchers to conduct meta-analyses in recent years, in order to clarify the potential effect of vitamin D supplementation on birth weight. However, we would like to highlight two randomized controlled trials (RCTs) that were not included in the systematic reviews and meta-analyses. We will discuss the results of these trials later. One such trial, which was recently opened, was conducted in 164 pairs of pregnant mothers and their infants at King Fahad Medical City in Riyadh, Saudi Arabia. The pregnant women in the study were randomized into two groups according to vitamin D supplementation: 400 IU (Group 1) versus 4000 IU (Group 2). The number of IUGRs was lower in Group 2 (9.6%) than in Group 1 (22.2%) [91]. Another randomized controlled trial involving 162 mother-infant pairs at a tertiary perinatal care center in Saudi Arabia investigated the efficacy and safety of prenatal vitamin D supplementation at 2000 and 4000 IU/day versus 400 IU/day. The results showed that mean serum calcidiol concentrations at delivery and in cord blood were significantly higher in the 2000 and 4000 IU groups than in the 400 IU/d group, with the highest concentrations observed in the 4000 IU/d group. There were no adverse events related to vitamin D supplementation. In short, vitamin D supplementation of 2000 and 4000 IU per day was safe during pregnancy. The 4000 IU per day dose was the most effective in optimizing serum calcidiol concentrations in mothers and their infants [88].

A systematic review and meta-analysis of randomized controlled trials (RCTs) showed that prenatal vitamin D supplementation was associated with increased maternal and cord serum calcidiol concentrations, increased mean birth weight, and a reduced risk of SGA. However, greater effects were not found consistently with higher effective doses. This is likely due to poor adherence to supplementation regimens or a lower vitamin D content than indicated on the supplement label. These factors reflect the low quality of many of the included trials [92]. Another systematic review and meta-analysis of RCTs concluded that vitamin D supplementation during pregnancy is safe and effective. It does not increase the risk of fetal or neonatal mortality or congenital abnormality. It reduces the risk of SGA and improves neonatal calcium levels, skinfold thickness, and fetal growth (greater weight and height at birth). Late supplementation (initiation at ≥20 weeks’ gestation) improved birth weight, whereas early supplementation (initiation at <20 weeks’ gestation) did not. However, this result should be interpreted with caution due to the heterogeneity of the included RCTs, as there were differences in the characteristics of the studied population, ethnicity, geographical conditions, and the timing and dose of vitamin D administered during pregnancy [93]. A 2019 systematic review and meta-analysis of RCTs assessed the effects of oral vitamin D supplementation during pregnancy on neonatal anthropometric measurements and the incidence of LBW or SGA. Daily doses ranged from 200 to 4000 IU, and single high-intensity intermittent interventions ranged from 35,000 to 600,000 IU. Compared to the control groups, the risk of SGA was lower in the intervention groups. Most of the included studies demonstrated the benefits of supplementation at or above the recommended dose of 600 IU per day as set out by the Institute of Medicine [87]. However, the meta-regression analysis did not reveal a dose-dependent effect of vitamin D on birth size, likely because of the limited number of studies included (only thirteen RCTs). In summary, the results of this systematic review and meta-analysis confirm that vitamin D is essential for fetal growth and development, and that it has well-established effects on birth size. Further research is needed to evaluate the dose-dependent effects of vitamin D alone or in combination with other micronutrients [94]. Based on strong evidence, the U.S. Food and Drug Administration (FDA) recently recommended pharmacological vitamin D supplementation during pregnancy of at least 2000–4000 IU per day to achieve blood vitamin D levels of at least 30 ng/mL, preferably 40 ng/mL. Additionally, the FDA established 4000 IU/day as the tolerable upper limit for pregnant women [95].

A quasi-experimental clinical trial conducted at the Duhok Maternity Hospital in Iraqi Kurdistan has recently been published. The objective was to investigate the effectiveness of combining vitamin D supplements with calcium compared to the use of vitamin D supplements alone on neonatal anthropometry. Pregnant women from 20 weeks of gestation until delivery were divided into two groups. One group received a daily dose of 1000 IU of vitamin D (n = 41), and the other received a daily dose of 1000 IU of vitamin D and 500 mg of calcium (n = 36). Newborns in the vitamin D + calcium group were less likely to have low birth weight (5.71% vs. 31.58%; *p* = 0.0066) or short birth length (5.71% vs. 44.74%; *p* = 0.0007) than those in the vitamin D group. Thus, this study provides evidence supporting the positive effects of daily co-supplementation of vitamin D and calcium during the third trimester of pregnancy on neonatal anthropometry [96].

Finally, it should be noted that vitamin D status could affect several aspects of health other than metabolism, including its anti-inflammatory and immune regulatory action. A clinical trial was designed to evaluate the efficacy of two doses of vitamin D supplementation during pregnancy on maternal and cord blood vitamin D status, inflammatory biomarkers, and neonatal size. A total of 84 pregnant women with a gestational age of less than 12 weeks were randomly assigned to one of two groups that received different doses of vitamin D supplementation: 1000 IU or 2000 IU per day. Maternal biochemical assessments (calcidiol, IL-1, IL-6, and TNF-α) were performed at the beginning of the study and at 34 weeks of gestation. After delivery, the same biochemical assessments were performed, as well as an evaluation of birth size in the newborn. The results concluded that 2000 IU/day of vitamin D supplementation from the first trimester of pregnancy was more effective than 1000 IU/day in improving the vitamin D status of pregnant women and decreasing inflammatory biomarkers (TNF-α in the mother and IL-6 in the cord blood). Furthermore, 2000 IU/day supplementation improved neonatal birth size (weight, length, and head circumference) compared to 1000 IU/day [97].

## 7. Conclusions

Vitamin D deficiency is a significant public health concern with numerous health consequences. Pregnant women and newborn babies are considered to be at high risk for vitamin D deficiency. During pregnancy, vitamin D is crucial for placental function, calcium homeostasis, and fetal bone mineralization, all of which are essential for fetal development and growth.

This review confirms the high prevalence of vitamin D deficiency in pregnant women and newborns worldwide, including in countries with high levels of sunshine. Furthermore, most studies indicate that fetal and/or neonatal vitamin D levels are directly related to maternal levels. Therefore, vitamin D deficiency in pregnant women may result in decreased calcium absorption and transfer to the placenta, as well as uteroplacental dysfunction. Monitoring vitamin D status in pregnant women and newborns should certainly be a global priority.

Several studies have shown an association between low maternal vitamin D levels and adverse fetal growth outcomes, such as IUGR. The relationship between maternal vitamin D status and neonatal anthropometry has been documented throughout pregnancy, though discrepancies have arisen due to heterogeneity in the design of the study. Therefore, monitoring vitamin D status during pregnancy is important, as deficiency at the beginning and end of pregnancy could affect fetal growth.

The biological basis of the association between maternal vitamin D status and fetal growth patterns is unclear. However, several concurrent pathophysiological mechanisms have been proposed to explain this association. For instance, the presence of VDR and CYP27B1 in early-pregnancy placentas suggests that vitamin D plays a significant role in implantation and placental function. Vitamin D deficiency could lead to abnormal placentation and cause pregnancy complications such as spontaneous abortion or intrauterine growth restriction. Conversely, the primary maternal-fetal flow of calcium and phosphate occurs through the placenta and fetal circulation to the bone. Vitamin D deficiency in pregnant women is likely to be accompanied by maternal hypocalcemia. If placental calcium transfer is deficient, compensatory fetal hyperparathyroidism leads to calcium reabsorption from the fetal skeleton, resulting in a decrease in bone mineral content and potentially causing intrauterine growth restriction. Additionally, experimental evidence suggests that gestational vitamin D deficiency causes placental insufficiency, uteroplacental dysfunction, and IUGR by inducing placental inflammation. Given its anti-inflammatory properties, adequate vitamin D (cholecalciferol) supplementation could potentially address the aforementioned pathophysiological mechanisms and prevent adverse pregnancy outcomes, particularly IUGR, in vitamin D-deficient pregnant women. Most authors suggest that vitamin D supplementation should begin during the preconception period, and that future research should investigate the mechanisms by which vitamin D contributes to fetal growth.

Although the results of the interventions are inconclusive owing to various reasons already mentioned, there is substantial evidence to suggest that maternal vitamin D supplementation (cholecalciferol) may improve maternal and neonatal outcomes. In other words, although vitamin D is essential for fetal development and growth, there is insufficient evidence to make recommendations regarding the optimal amount of prenatal vitamin D supplementation to reduce the risk of IUGR. A coordinated effort to conduct large-scale randomized controlled trials investigating the dose-dependent effect of prenatal vitamin D supplementation on fetal growth patterns should be an immediate priority.

## Figures and Tables

**Figure 2 ijms-26-11422-f002:**
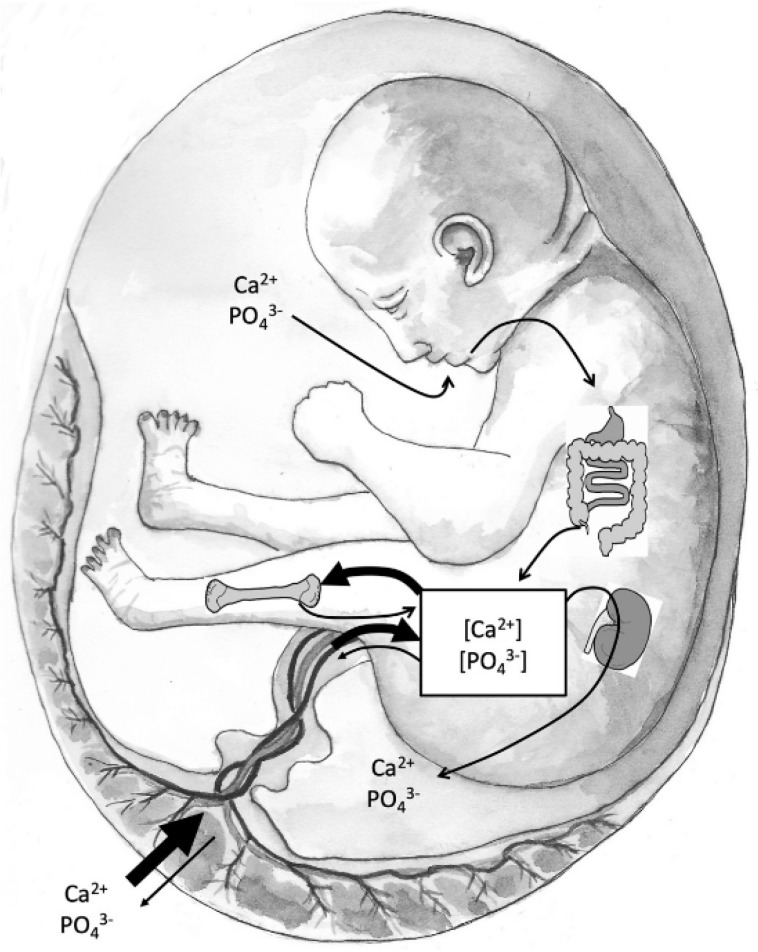
Sources of minerals during fetal development (reproduced with permission from Ryan et al. [37]). The main flux of calcium and phosphate is across the placenta and through the fetal circulation into bone; however, some mineral returns to the maternal circulation (backflux). The fetal kidneys filter the blood with little or no active reabsorption of minerals, such that the excretion of calcium is proportionate to its serum concentrations. Amniotic fluid, which is largely composed of fetal urine, is swallowed and absorbed, thereby restoring minerals to the circulation. However, this renal-amniotic-intestinal loop is likely a minor component for fetal mineral homeostasis.

**Table 1 ijms-26-11422-t001:** American Endocrine Society criteria for classification of vitamin D status.

Vitamin D Status	Calcidiol Level (1 ng/mL Corresponds to 2.5 nmol/L)
Vitamin D deficiency	<20 ng/m (<50 nmol/L)
Vitamin D insufficiency	20 to 29 ng/mL (51–74 nmol/L)
Vitamin D sufficiency	≥30 ng/m (≥75 nmol/L)

**Table 2 ijms-26-11422-t002:** Prevalence of vitamin D deficiency in the pregnant women.

Author and Year	Region/Country (Latitude)	Calcidiol < 20 ng/mL (%)
Woo et al., 2023 [48]	Columbus, OH, USA (39°57′ N) Detroit, MI, USA (42°19′ N)	42
Stoica et al., 2024 [49]	Târgu Mures, Romania (46°32′ N)	74
Jaiswal et al., 2024 [50]	Barabanki, India (26°55′ N)	66
Kokkinari et al., 2024 [51]	Athens, Greece (37°59′ N)	58
Dragomir et al., 2024 [52]	Bucharest, Romania (44°24′ N)	45
Singh et al., 2024 [53]	Jhansi, India (25°26′55″ N)	55
O’Callaghan et al., 2024 [54]	Various intercity regions, UK	67
Reverzani et al., 2025 [55]	Kampala, Uganda (0°18′ N)	40
Saccone et al., 2025 [56]	Naples, Italy (40°50′ N)	56
Gironés Soriano et al., 2025 [57]	Sagunto, Spain (39°40′ N)	53

## Data Availability

No new data were created or analyzed in this study. Data sharing is not applicable to this article.

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
