# Peer review of "Vitamin D and Intrauterine Growth Restriction (IUGR)"

_ijms, 2025, doi:10.3390/ijms262311422_

Round 1
Reviewer 1 Report
Comments and Suggestions for Authors
The narrative review from Durá-Travé and Gallinas-Victoriano is clearly organized and deals with different important issues related to the vitamin D status in pregnancy. These include the prevalence of vitamin d deficiency in pregnant women, and the relationships between maternal vitamin D status and neonatal vitamin d statuts as well as well as fetal growth patterns. It also describes studies of vitamin d supplementation during pregnancy.
My main concerns are about tables, figures and references.
Tables
- Table 1 reports the American Endocrine Society criteria for classification of vitamin D status, defined as “the most widely accepted and used by the authors”. The choice of the authors is legitimate, but the debate in the literature on thresholds of vitamin D sufficiency/deficiency is open and should be reported (see, just as a few examples: Bouillon R. Nat Rev Endocrinol. 2017 doi: 10.1038/nrendo.2017.31; Giustina et al, Endocr Rev. 2024 doi: 10.1210/endrev/bnae009; Giustina et al, Nat Rev Endocrinol https://doi.org/10.1038/s41574-023-00942-0).
- Table 2 is mainly based on a single reference dating 2016 (almost 10 years ago), with the addition of data of vitamin deficiency in Africa extrapolated from a more recent (and more accurate and structured) review. Other data are discussed in the text. I suggest making the effort to create an original table that includes all (or most) of the more recent data. In addition, in its actual form there is an error in the table legend. The author state “Prevalence of maternal and newborn vitamin D deficiency [calcidiol <30 ng/m L]” : but the two refs from which data are derived both use cut-off of <20 ng/m L.
Minor points for Table 2:
- it mis-cited as Table 1 in paragraph 3;
- it should be moved to paragraph 3
Figures
The three figures are copied (and not adapted) from articles that authors did not co-authored. Furthermore, it is not clear the rationale behind the “pick up” of those particular figures from the set of papers relating to the specific topic they should illustrate. Let's take Figure 1 as an example. Why the authors choose this particular study among the many cited and discussed?
In my opinion, such figures extrapolated from single papers are not appropriate nor general enough for a narrative review. This applies to all three figures.
I suggest instead elaborating original schemes/models to illustrate the concepts and/or data that the authors consider most important.
References:
I must say that evaluating the review was made particularly difficult by errors in the bibliography:
- There is a shift in ref numbers (for sure from ref 35 on, but check also before): ref 35 should be 36 and so on until the end – last citation in the text is ref 93, last ref in the reference list is 94
- Even taking into account the shift, some references are not corresponding to the citation in the text: for example, the observational study conducted at the Obstetrics Clinic of the County Clinical Hospital in Târgu Mureș, Romania (discussed in lanes 234-241) is referred to as ref 54, but neither ref 54 nor ref 55 appear to refer to such a study.
- lanes 343-356: the reference of the animal study is missing
Please carefully check all references and their correspondence with the reference list.
Author Response
Responses to reviewer-1
First of all, we would like to thank you for your suggestions regarding this article
NOTE: The corrected text of the new version is in red
Comments and Suggestions for Authors
The narrative review from Durá-Travé and Gallinas-Victoriano is clearly organized and deals with different important issues related to the vitamin D status in pregnancy. These include the prevalence of vitamin d deficiency in pregnant women, and the relationships between maternal vitamin D status and neonatal vitamin d statuts as well as well as fetal growth patterns. It also describes studies of vitamin d supplementation during pregnancy.
My main concerns are about tables, figures and references.
Tables
-
Table 1 reports the American Endocrine Society criteria for classification of vitamin D status, defined as “the most widely accepted and used by the authors”. The choice of the authors is legitimate, but the debate in the literature on thresholds of vitamin D sufficiency/deficiency is open and should be reported (see, just as a few examples: Bouillon R. Nat Rev Endocrinol. 2017 doi: 10.1038/nrendo.2017.31; Giustina et al, Endocr Rev. 2024 doi: 10.1210/endrev/bnae009; Giustina et al, Nat Rev Endocrinol https://doi.org/10.1038/s41574-023-00942-0).
Indeed, it is a widely debated topic that we have not wished to delve into in depth as it is not the subject of this review.
Even though it is generally accepted that serum calcidiol levels are the best indicator of body vitamin D stores, there is some controversy with regard to the limits that define normality. The American Institute of Medicine proposed considering calcidiol levels of 20 ng/mL as the threshold of normality for the organic vitamin D content, given the practical absence of significant metabolic alterations at concentrations above 20 ng/mL, while levels below these figures induce an increase in PTH and bone alterations might be already detected. However, subsequently, the US Endocrine Society considered it prudential, in order to safeguard the multiple extra-skeletal functions of vitamin D, to modify the calcidiol values defining vitamin D sufficiency without modifying the already established cut-off point for vitamin D deficiency but intercalating between these two states the concept of vitamin D insufficiency, which should also be overcome
The article by Giustina et al. (Consensus statement on vitamin D status assessment and supplementation: why, when, and how. Endocr Rev. 2024; 45: 625–654) is extraordinary. It summarizes what was discussed at the 6th International Conference “Controversies in Vitamin D” (Florence, Italy, 2022). However, their conclusions note that while calcidiol is currently the most accepted biomarker for assessing vitamin D status, its optimal levels remain a matter of debate.
-
Table 2 is mainly based on a single reference dating 2016 (almost 10 years ago), with the addition of data of vitamin deficiency in Africa extrapolated from a more recent (and more accurate and structured) review. Other data are discussed in the text. I suggest making the effort to create an original table that includes all (or most) of the more recent data. In addition, in its actual form there is an error in the table legend. The author state “Prevalence of maternal and newborn vitamin D deficiency [calcidiol <30 ng/m L]” : but the two refs from which data are derived both use cut-off of <20 ng/m L.
Minor points for Table 2:
-
it mis-cited as Table 1 in paragraph 3;
-
it should be moved to paragraph 3
Numerous local studies have subsequently been published, that is, in different countries, but without global characteristics. However, the data presented in Table 2 represent a global summary of maternal and neonatal vitamin D status involving pregnant women and their newborns from different population groups corresponding to WHO regions from which data were available (Saraf et al. [15]). Table 2 is completed with the corresponding data from the African continent (Mogire et al. [43]). Throughout this paper, multiple references are made to local studies conducted in countries on various continents, the results of which practically corroborate the data presented in Table 2.
Previous table... “[calcidiol <30 ng/m L]”
has been changed to... “[calcidiol <20 ng/m L]”
The minor points for Table 2 have been corrected.
Figures
The three figures are copied (and not adapted) from articles that authors did not co-authored. Furthermore, it is not clear the rationale behind the “pick up” of those particular figures from the set of papers relating to the specific topic they should illustrate. Let's take Figure 1 as an example. Why the authors choose this particular study among the many cited and discussed?
In my opinion, such figures extrapolated from single papers are not appropriate nor general enough for a narrative review. This applies to all three figures.
I suggest instead elaborating original schemes/models to illustrate the concepts and/or data that the authors consider most important.
We consider this comment very pertinent. Furthermore, Figure 1 was repeated on lines 258-261 and Figure 3 on lines 403-408 of the previous version, so they have been removed. However, we believe Figure 2 (now Figure 1) should be retained, as, together with the text, it allows us to understand part of maternal-fetal adaptive physiology.
References:
I must say that evaluating the review was made particularly difficult by errors in the bibliography:
-
There is a shift in ref numbers (for sure from ref 35 on, but check also before): ref 35 should be 36 and so on until the end – last citation in the text is ref 93, last ref in the reference list is 94
-
Even taking into account the shift, some references are not corresponding to the citation in the text: for example, the observational study conducted at the Obstetrics Clinic of the County Clinical Hospital in Târgu Mureș, Romania (discussed in lanes 234-241) is referred to as ref 54, but neither ref 54 nor ref 55 appear to refer to such a study.
-
lanes 343-356: the reference of the animal study is missing
Please carefully check all references and their correspondence with the reference list.
We sincerely regret the inconsistency in the reference list, which obviously makes reviewing the article difficult. This inconsistency has been reviewed and corrected.
We would like to express our thanks to referee for your suggestions and positive criticisms.
We hope every made question have been answered adequately.
Yours sincerely,
Teodoro Durá-Travé
Reviewer 2 Report
Comments and Suggestions for Authors
In present work, Durá-Travé and Gallinas-Victoriano try to review the possible influence of vitamin D deficiency on the pathogenetic mechanisms of intrauterine growth restriction, and the potential benefits of vitamin D supplementation during pregnancy on fetal anthropometry. However, there are some questions that should be answered.
Major concerns
- Lines 66-67, ‘covering publications from January 2014 to December 2024’. However, many publications cited by this manuscript are before 2014. In addition, some publications very related with this manuscript are not included. For example,
Wang J, Qiu F, Zhao Y, Gu S, Wang J, Zhang H. Exploration of fetal growth restriction induced by vitamin D deficiency in rats via Hippo-YAP signaling pathway. Placenta. 2022;128:91-99.
- A color figure about vitamin D metabolism and adaptive changes during pregnancy should be added, including many cartoon images (pregnant female, skin, fetus liver, kidney, and placenta).
- Figure 2 should be revised based on your review with different colors. In addition, figure legend is needed.
- Title should be revised. "Maternal Vitamin D Status, Supplementation, and the Risk of Intrauterine Growth Restriction" may be suitable.
- In the conclusion section, future research directions should be added.
- English writing should be checked and revised throughout the manuscript. There are many low wrongs.
Minor concerns
- Line 3, delete two ‘MD’.
- Keywords should be revised, for example, low birth weight; small for gestational age; Vitamin D status; vitamin D deficiency; vitamin D supplementation.
- Lines 106-139, too long paragraph.
- Line 139, correct ‘fetus. [2, 38].’.
- Line 150, correct ‘factor-alfa’.
- Table 2 is repeated with Lines 185-191. Table 2 should be deleted.
- Figure 1 is repeated with Lines 258-261. Figure 1 should be deleted.
- Figure 3 is repeated with Lines 403-408. Figure 3 should be deleted.
- Line 492, correct ‘[90).’.
- Line 504, correct ‘P = 0.0066’, ‘P’ should be italic. Please check and revise it throughout the manuscript.
- Line 523, correct ‘Conclusions.’.
- Conclusions should be refined, including one or two paragraphs.
- Reference section. Inconsistent reference format.
For example, Ref. 1, ‘Wacker M, Holick MF’, but Ref. 2, ‘Hossein-Nezhad, A.; Holick, M.F.’.
Ref. 4, correct ‘problem?.’.
The format of references should be based on MDPI style.
Author 1, A.B.; Author 2, C.D. Title of the article. Abbreviated Journal Name Year, Volume, page range.
Comments on the Quality of English LanguageThe English could be improved to more clearly express the research.
Author Response
Responses to reviewer-2
First of all, we would like to thank you for your suggestions regarding this article
NOTE: The corrected text of the new version is in red
Comments and Suggestions for Authors
In present work, Durá-Travé and Gallinas-Victoriano try to review the possible influence of vitamin D deficiency on the pathogenetic mechanisms of intrauterine growth restriction, and the potential benefits of vitamin D supplementation during pregnancy on fetal anthropometry. However, there are some questions that should be answered.
Major concerns
-
Lines 66-67, ‘covering publications from January 2014 to December 2024’. However, many publications cited by this manuscript are before 2014. In addition, some publications very related with this manuscript are not included. For example:
Wang J, Qiu F, Zhao Y, Gu S, Wang J, Zhang H. Exploration of fetal growth restriction induced by vitamin D deficiency in rats via Hippo-YAP signaling pathway. Placenta. 2022;128:91-99.
Indeed, among the 94 citations in the article, 13 papers were consulted, corresponding to the years 2010 (two), 2011 (two), 2012 (one), and 2013 (seven), whose contribution we considered relevant to the present paper. There is also a citation corresponding to the year 2025. Therefore, line 70 (second version) will now read: "...covered publications from January 2010 to June 2025."
In addition, a citation from 2007 (Clayton et al.) that defined the criteria of the SGA and IUGR has been removed, considering them a globally accepted concept.
The publication suggested by the authors seems very appropriate to us and has been included in the text (lines: 370-376):
“More recently, an experimental study in rats showed that IUGR in pregnant rats with vitamin D deficiency was significant. Placental structure and function were impaired and there was a clear inflammatory response accompanied by a significant increase in the phosphorylation level of the transcriptional coactivator Yes-associated protein (YAP). In other words, maternal vitamin D deficiency leads to placental dysplasia and IUGR, and these abnormal changes may be linked to the activation of the Hippo-YAP signalling pathway [74].”.
-
A color figure about vitamin D metabolism and adaptive changes during pregnancy should be added, including many cartoon images (pregnant female, skin, fetus liver, kidney, and placenta).
After additional evaluation, we consider that it would not be necessary to add a figure of such characteristics, since we understand that the text is sufficiently explicit on this subject.
-
Figure 2 should be revised based on your review with different colors. In addition, figure legend is needed.
Figure 2 (now Figure 1) is the original that has been kindly provided to us by its authors (Ryan et al., 2020), and we think that, together with the ad hoc text (see below), it is well understood.
Lines 334-339: “Figure 1 shows the sources of minerals during fetal development. The main flow of calcium and phosphate occurs through the placenta and fetal circulation to the bone. The fetal kidneys filter blood with little active mineral reabsorption. The amniotic fluid, which is composed primarily of fetal urine, is swallowed and absorbed. However, the renal-amniotic-intestinal pathway plays a minor role in fetal mineral homeostasis because the placenta is the main supplier of minerals.”
-
Title should be revised. "Maternal Vitamin D Status, Supplementation, and the Risk of Intrauterine Growth Restriction" may be suitable.
At the reviewer's suggestion, title 5 has been modified.
Previous text... “Maternal Vitamin D Status and Fetal Growth Patterns”
has been changed to... "Maternal Vitamin D Status and Risk of Intrauterine Growth Restriction"
-
In the conclusion section, future research directions should be added.
The last paragraph of the conclusions mentions a suggestion that was practically unanimous among the different authors:
““...although vitamin D is essential for fetal development and growth, there is insufficient evidence to make recommendations for the optimal amount of prenatal vitamin D supplementation to reduce the risk of IUGR. A coordinated effort to conduct large-scale randomised controlled trials investigating the dose-dependent effect of prenatal vitamin D supplementation on fetal growth patterns should be an immediate priority”.
-
English writing should be checked and revised throughout the manuscript. There are many low wrongs.
The text has been revised and corrected
Minor concerns
-
Line 3, delete two ‘MD’.
Both MDs have been removed from the text
-
Keywords should be revised, for example, low birth weight; small for gestational age; Vitamin D status; vitamin D deficiency; vitamin D supplementation.
-
A new keyword has been added: pregnancy
-
Lines 106-139, too long paragraph.
For better follow-up and understanding, this very long paragraph has been divided into four fragments.
-
Line 139, correct ‘fetus. [2, 38].”
For better understanding the sentence has been modified:
Previous text... “This epigenetic imbalance in the vitamin D feedback loop greatly contributes to increased maternal calcitriol levels during pregnancy and ensures greater calcium transfer between mother and fetus [2, 38].”
(now lines 144-146) has been changed to... “This epigenetic imbalance of vitamin D feedback catabolite would greatly contribute to the increase in maternal calcitriol levels during pregnancy. This ensures greater maternal-fetal calcium transfer [2, 39].”
-
Line 150, correct ‘factor-alfa’.
Previous text... “tumor necrosis factor alfa (TNF-α)”
(now line 158) has been changed to...”tumor necrosis factor alpha (TNF-α)”
-
Table 2 is repeated with Lines 185-191. Table 2 should be deleted.
Although it could be argued that data is repeated, we understand that Table 2 facilitates the understanding of the text since it allows the referenced data to be quickly captured.
-
Figure 1 is repeated with Lines 258-261. Figure 1 should be deleted.
Figure 1 has been removed
-
Figure 3 is repeated with Lines 403-408. Figure 3 should be deleted.
Figure 3 has been removed
-
Line 492, correct “[90)”
Previous text... [90).
(now line 514) has been changed to... [91].
-
Line 504, correct ‘P = 0.0066’, ‘P’ should be italic. Please check and revise it throughout the manuscript.
Line 249: (r = 0.96, p < 0.01)
Line 256: (r = 0.68; p < 0.0001).
Line 527 (before line 504): (5.71% vs. 31.58%; P = 0.0066) or short birth length (5.71% vs. 44.74%; P = 0.0007).
-
Line 523, correct ‘Conclusions.’.
Previous text... “7. Conclusiones”
has been changed to… “7. Conclusions”
-
Conclusions should be refined, including one or two paragraphs.
The heterogeneity of the many articles reviewed (more than those referenced) makes it very difficult to write brief conclusions, as important elements could be overlooked. Therefore, we have been forced to write relatively long conclusions, but they clarify the current situation on the topic at hand.
-
Reference section. Inconsistent reference format.
For example, Ref. 1, ‘Wacker M, Holick MF’, but Ref. 2, ‘Hossein-Nezhad, A.; Holick, M.F.’. Ref. 4, correct ‘problem?.’.
The format of references should be based on MDPI style: Author 1, A.B.; Author 2, C.D. Title of the article. Abbreviated Journal Name Year, Volume, page range.
The format of the references has been adapted to the MDPI style
We would like to express our thanks to referee for your suggestions and positive criticisms.
We hope every made question have been answered adequately.
Yours sincerely,
Teodoro Durá-Travé
Round 2
Reviewer 1 Report
Comments and Suggestions for Authors
I thank the authors for the answers, but unfortunately most of my points remained unresolved.
Concerning the first point, related to table 1: indeed, I did not ask to delve into depth of the topic, but just to mention the fact that a debate is ongoing and there is no consensus about optimal levels. It is important to mention the issue (with adequate reference(s)) in the text.
Concerning table 2, my main concern remains unanswered: the table refers - with the exception of data relating to Africa - to data reviewed ten years ago. Such data, moreover, are quickly accessible in the abstract of the cited review (Saraf et al. [15]). I don't see how this table adds value to this review. A table summarizing up-to-date data, even if more geographically fragmented, would be much more useful for the reader.
Regarding the figures, I appreciate the removal of the two inappropriate ones, although I noticed the authors didn't consider the suggestion to develop original one(s). I understand that developing original illustrations takes time, but, while not mandatory, figures can help the reader and improve the quality of the review.
Finally, minor errors in the reference list are still present:
- Kovacs, C.S. Maternal mineral and bone metabolism during pregnancy, lactation, and post-weaning recovery. Physiol. Rev. 2016, 96, 449–547: same paper cited as ref 35 and 37
- ref 00 (?) in table 2 legend
Author Response
Responses to reviewer-1 (second round)
First of all, we would like to thank you for your suggestions regarding this article
NOTE: The corrected text of the new version is in red
Comments and Suggestions for Authors
I thank the authors for the answers, but unfortunately most of my points remained unresolved.
Concerning the first point, related to table 1: indeed, I did not ask to delve into depth of the topic, but just to mention the fact that a debate is ongoing and there is no consensus about optimal levels. It is important to mention the issue (with adequate reference(s)) in the text.
In the new version the following sentence has been added (lines 41-43): “However, while calcidiol is currently the most accepted biomarker for assessing vitamin D status, its optimal levels remain a matter of debate”. The reference from Giustina et al, 2024 (doi: 10.1210/endrev/bnae009) that you suggested us has also been included.
Concerning table 2, my main concern remains unanswered: the table refers - with the exception of data relating to Africa - to data reviewed ten years ago. Such data, moreover, are quickly accessible in the abstract of the cited review (Saraf et al. [15]). I don't see how this table adds value to this review. A table summarizing up-to-date data, even if more geographically fragmented, would be much more useful for the reader.
We appreciate your suggestion. Table 2 has been removed from the previous version and replaced with a table summarising the latest data on the prevalence of vitamin D deficiency.
The following sentence has been added to the new version (lines 248-250): “More recently, several observational studies have been conducted concerning the prevalence of vitamin D deficiency in pregnant women in countries with diverse ethnic and geographic characteristics.Table 2 shows the results of some of these studies.”
Regarding the figures, I appreciate the removal of the two inappropriate ones, although I noticed the authors didn't consider the suggestion to develop original one(s). I understand that developing original illustrations takes time, but, while not mandatory, figures can help the reader and improve the quality of the review.
Finally, minor errors in the reference list are still present:
-
Kovacs, C.S. Maternal mineral and bone metabolism during pregnancy, lactation, and post-weaning recovery. Physiol. Rev. 2016, 96, 449–547: same paper cited as ref 35 and 37
-
ref 00 (?) in table 2 legend
We apologize for this transcription error. The reference list has been revised and the errors noted have been corrected.
We hope every made question have been answered adequately.
Yours sincerely,
Teodoro Durá-Travé
Reviewer 2 Report
Comments and Suggestions for Authors
Thanks for author’s responses. However, there are STILL some questions that should be answered.
- A color figure about vitamin D metabolism and adaptive changes during pregnancy should be added, including many cartoon images (pregnant female, skin, fetus liver, kidney, and placenta).
The authors respond that ‘After additional evaluation, we consider that it would not be necessary to add a figure of such characteristics, since we understand that the text is sufficiently explicit on this subject’.
This reviewer does not agree with authors’ idea. For a higher quality review, several colour figures are necessary. This color figure is needed.
- Figure 2 is provided to us by its authors (Ryan et al., 2020), do the authors obtain permission from Ryan et al. In addition, figure legend is needed. Only explaining in the text is not enough.
- English writing should be checked and revised throughout the manuscript. There are many low wrongs.
For example,
Line 1, correct ‘—Review’.
Keywords should be revised, some are not Keywords. ‘birth weight’ is a Keyword, but ‘low birth weight’ is not. ‘gestational age’ is a Keyword, but ‘small for gestational age’ is not. ‘Vitamin D’ is a Keyword, but ‘Vitamin D status; vitamin D deficiency; vitamin D supplementation’ are not. There are two ‘pregnancy’.
Line 158, delete (IFN-γ). If the abbreviation only appears once, please delete it, including the List of abbreviation. Please check abbreviation one by one.
Line 256, ‘p < 0.0001’, Line 527, ‘P = 0.0066’ and ‘P = 0.0007’. ‘p’ or ‘P’? In addition, the number should be not italic.
Line 546, correct ‘Conclusions.’ to ‘Conclusions’.
The format of references should be based on MDPI style.
Author 1, A.B.; Author 2, C.D. Title of the article. Abbreviated Journal Name Year, Volume, page range.
Line 645, correct ‘Ann. NY Acad. Sci.’ to ‘Ann. N. Y. Acad. Sci.’.
Line 812, correct ‘J Clin Endocrinol Metab’ to ‘J. Clin. Endocrinol. Metab.’.
Please check these one by one.
Comments on the Quality of English LanguageThe English could be improved to more clearly express the research.
Author Response
Responses to reviewer-2 (second round)
First of all, we would like to thank you for your suggestions regarding this article
NOTE: The corrected text of the new version is in red
Comments and Suggestions for Authors
Thanks for author’s responses. However, there are STILL some questions that should be answered.
-
A color figure about vitamin D metabolism and adaptive changes during pregnancy should be added, including many cartoon images (pregnant female, skin, fetus liver, kidney, and placenta).
The authors respond that ‘After additional evaluation, we consider that it would not be necessary to add a figure of such characteristics, since we understand that the text is sufficiently explicit on this subject’.
This reviewer does not agree with authors’ idea. For a higher quality review, several colour figures are necessary. This color figure is needed.
A color figure on vitamin D metabolism and adaptive changes during pregnancy (Figure 1) with its corresponding legend has been added.
-
Figure 2 is provided to us by its authors (Ryan et al., 2020), do the authors obtain permission from Ryan et al. In addition, figure legend is needed. Only explaining in the text is not enough.
The legend corresponding to Figure 2 has been added (its content has been removed from the text)
-
English writing should be checked and revised throughout the manuscript. There are many low wrongs.
For example,
Line 1, correct ‘—Review’.
‘Review’ is a note from the publisher
Keywords should be revised, some are not Keywords. ‘birth weight’ is a Keyword, but ‘low birth weight’ is not. ‘gestational age’ is a Keyword, but ‘small for gestational age’ is not. ‘
’ is a Keyword, but ‘Vitamin D status; vitamin D deficiency; vitamin D supplementation’ are not. There are two ‘pregnancy’.
In the PubMed database of the US National Library of Medicine, the following are listed as Medical Subject Headings or keywords: 'vitamin D', 'vitamin D deficiency', 'small for gestational age', 'low birth weight', and 'vitamin D supplementation', which facilitated our bibliographic search. In fact, these Medical Subject Headings or keywords have been widely used in the consulted bibliography. However, 'vitamin D status' is not listed as a Medical Subject Heading or keyword and has been removed. The duplication of 'pregnancy' has been corrected.
Line 158, delete (IFN-γ). If the abbreviation only appears once, please delete it, including the List of abbreviation. Please check abbreviation one by one.
All abbreviations have been reviewed and those that are not repeated throughout the text have been removed, such as MARRS (line 108), IFN-γ (line 176), and TGF-β (line 177).
Line 256, ‘p < 0.0001’, Line 527, ‘P = 0.0066’ and ‘P = 0.0007’. ‘p’ or ‘P’? In addition, the number should be not italic.
To standardize the text, it now presents:
Líne 269: (r = 0.96, p < 0.01)
Líne 276: (r = 0.68; p < 0.0001).
Líne 549: (5.71% vs. 31.58%; p = 0.0066) or short birth length (5.71% vs. 44.74%; p = 0.0007).
Line 546, correct ‘Conclusions.’ to ‘Conclusions’.
Now it says: Conclusions (line 568)
The format of references should be based on MDPI style.
Author 1, A.B.; Author 2, C.D. Title of the article. Abbreviated Journal Name Year, Volume, page range.
Line 645, correct ‘Ann. NY Acad. Sci.’ to ‘Ann. N. Y. Acad. Sci.’.
Line 812, correct ‘J Clin Endocrinol Metab’ to ‘J. Clin. Endocrinol. Metab.’.
The format of the references has been reviewed and corrected according to the MDPI style.
Please check these one by one.
The English could be improved to more clearly express the research.
The text has been reviewed and any spelling errors found have been corrected.
We hope every made question have been answered adequately.
Yours sincerely,
Teodoro Durá-Travé
Round 3
Reviewer 1 Report
Comments and Suggestions for Authors
I thank the authors for the addition of the sentence on lines 41-43.
Concerning Table 2: the rationale of the choice of papers included is not clear, and only data relative to prevalence in mothers is reported, while the original one was comparing prevalence in mothers and in newborns. Furthermore, the table contains unacceptable errors. First of all, in the second column “Region/Country”, in two cases (ref. 48 and ref. 54), the Region/Country of affiliation of the authors has been reported instead of the Region/Country in which the study was conducted, leaving serious doubts about the method of data extraction for the compilation of the table. Additionally, in two other cases (ref. 52 and ref. 55) errors in the % of prevalence of vitamin D deficiency reported are present: for example, according to Dragomir et al. the prevalence of deficiency (i.e. <20 ng/ml) is 9.25 + 36.15 = 45.4%. The reported 73% also includes subjects with 25(OH)D levels between 21–29 ng/mL. Finally, Reverzani et al. reported a prevalence of 40%, while the authors reported a 54%.
For all these reasons, I kindly ask to remove the table.
Author Response
Responses to reviewer-1 (Round 3)
Comments and Suggestions for Authors
The English could be improved to more clearly express the research.
The English has been reviewed and corrected to improve the understanding of the article.
NOTE: The corrected text of the new version is in red
Yours sincerely,
Teodoro Durá-Travé
Reviewer 2 Report
Comments and Suggestions for Authors
Thanks for author’s responses. However, there are STILL some questions that should be answered.
- Title, delete ‘(IUGR)’.
- Percent match rate of this manuscript is 50%, which should be reduced.
- Figure 1, clarity and quality needs to be improved.
- Line 342, correct ‘figure 2’ to ‘Figure 2’.
- Line 374, please explain ‘KC’.
- Line 399, check ‘GraviD’.
- Conclusions section is too long, which should be refined.
- Line 790, correct ‘2022;128:91-99’.
The English could be improved to more clearly express the research.
Author Response
Responses to reviewer-2 (Round 3)
Comments and Suggestions for Authors
The English could be improved to more clearly express the research.
The English has been reviewed and corrected to improve the understanding of the article.
NOTE: The corrected text of the new version is in red
Yours sincerely,
Teodoro Durá-Travé